# The effectiveness of the buddy program training module to enhance the daily living function, social participation and emotional status of older adults in residential aged care homes

**Siti Noraini Asmuri**[1,2]*, **Masne Kadar**[2], **Nor Afifi Razaob**[2], **Chai Siaw Chui**[2], **Hanif Farhan Mohd Rasdi**[2]

**1** Faculty of Medicine and Health Sciences, Department of Rehabilitation Medicine, Universiti Putra Malaysia, Seri Kembangan, Malaysia, **2** Faculty of Health Sciences, Occupational Therapy Program, Centre for Rehabilitation and Special Needs Studies, Universiti Kebangsaan Malaysia, Bangi, Malaysia

* ctnoraini@upm.edu.my

## Abstract

### Background

The Compeer Model, which was originally designed to match individuals recovering from mental illness with volunteers from their community, served as the basis for the development of the buddy program. However, limited research was available related to the buddy program among older adults in a Malaysian context.

### Aim

The study aimed to identify the effectiveness of the buddy program training module to enhance the daily living function, social participation and emotional status of older adults in residential aged care homes.

### Methods

A quasi-experimental study was conducted with 30 pairs of buddies and older adults for both the experimental group and control group in two randomly selected residential aged care homes. The buddies in the experimental group received the buddy program training module related to activities of daily living (basic and instrumental) while the buddy-older adults pairs in the control group continued to perform their usual daily life activities in residential aged care homes. Baselines were performed before intervention and at eight weeks post-intervention.

### Results

Over the eight weeks, for the older adults in the experimental group, there was a significant main effect of time after the intervention on BADL ($p = 0.010$). There were no significant

Committee (contact via sepukm@ukm.edu.my) for researchers who meet the criteria for access to confidential data.

**Funding:** The study was supported by the Ministry of Education, Malaysia under the Fundamental Research Grant Scheme [FRGS/1/2019/SS06/UKM/02/8]. The funders had no role in study design, data collection and analysis, decision to publish, or preparation of the manuscript.

**Competing interests:** The authors have declared that no competing interests exist.

interaction effects for the experiment group and control group on IADL and social participation. Also, there were no significant interaction effects for all domains in emotional status: depression, anxiety and stress. For buddies, there was a significant interaction effect for depression ($p = 0.045$) in the control group.

## Conclusions

The buddy program training module can be used as a guideline for older adults with more significant disabilities in residential aged care homes in managing activities of daily living. Future studies could be implemented to explore the intergenerational buddy program among older adults and young children in the community.

## Introduction

The growing number of elderly individuals in Malaysia has resulted in a significant rise in the demand for specialized facilities required to maintain healthy lifestyle [1]. However, the current provision of facilities in aged care homes is inadequate to meet the anticipated surge in residents. This includes facilities available around the nursing home, facilities to engage in daily activities, and so on. Recently, a new scenario has developed where many children have sent their parents to aged care homes due to time constraints flowing from their career demands [2]. This scenario could lead to a lack of attention paid to parents and the occurrence of physical, emotional, financial, and material neglect [3], however, at present, older adults need comprehensive care encompassing their physical, social and psychological well-being [4].

Disability refers to the interaction between the individual and the environment that affects how a person carries out everyday activities, for example, limitations in daily activities and social involvement [5]. In Malaysia, Act 685, which is the Persons with Disabilities Act 2008, defines a person with a disability as an individual who has a long-term physical, mental, intellectual or sensory impairment that interacts with various limitations that may affect participation in the community [6]. People with disabilities often require support and may have a greater need for assistance with household tasks, as well as increased levels of care if they reside in residential aged care homes or hospitals [7].

There are various daily activities carried out by older adults in residential aged care homes and also in the community. These daily activities can be divided into two categories: namely, (1) Basic daily activities/personal daily activities BADL) and (2) instrumental daily activities (IADL) [8]. Daily activities, or Activities of Daily Living (ADL), are personal care activities including bathing, dressing, feeding, mobility, etc. [9]. This activity includes tasks that each individual performs starting from waking up, moving from one place to another, and going back to sleep at night. These activities are also known as basic daily activities (Basic Activities of Daily Living) as well as personal daily activities (Personal Activities of Daily Living). Meanwhile, instrumental daily activities or Instrumental Activities of Daily Living (IADL) are more complex activities that support daily life within the home and community environment [10]. Examples of IADL are medication care, telephone use, heavy housework, light housework, food preparation, etc. [11]. Social participation for older adults refers to a range of activities to promote the active engagement of older individuals in nursing homes, communities and various events that involve their family, friends and other individuals. The capacity to independently perform daily life activities and actively participate in

social and community endeavors has the potential to impact the emotional well-being and overall state of elderly individuals [12]. Emotions are strong feelings of the soul or a state of feeling or mind that is aroused and involves psychological and physiological changes in a person. These include the emotional stability of older adults such as managing anger, affection, tension, anxiety, etc. [13].

Prevalence studies showed that approximately one-third of older adults (34.6%) exhibit dependence in basic activities of daily living (BADL), while over half of the older adult population (53.5%) are classified as having limitations in instrumental activities of daily living (IADL) [14]. The prevalence rate for older adults' dependency in BADL showed that as many as 24.2% of the older adults reported that they needed help with at least one of the 10 items in the Modified Barthel Index (MBI) score [7]. These findings indicated that older adults require assistance with their daily routines, particularly with BADL and IADL. ADL scores are associated with comorbid conditions in older adults, and the association between ADL and mortality is higher for older adults residing in residential aged care homes compared to older adults living in the community [15].

Most of the interventions carried out on older adults predominantly emphasized group therapy activities rather than individualized approaches. A buddy program is an informal support system involving two individuals with the same purpose who mutually assist each other with their respective tasks. The relationships formed within this support network possess a therapeutic quality, where individuals assigned with responsibilities can acquire essential benefits such as support. Simultaneously, those who may be mentally and physically dependent can also derive advantages from participating in this mentoring program. This buddy program involved two individuals, 'Buddy Aids' and 'Buddy'. The 'Buddy Aid', otherwise known as a helper or guide, is an individual who is assigned the task and responsibility of assisting the 'Buddy'; i.e., the person who is guided or assisted. In previous studies, these buddy aids consisted of older adults [16], neighbors or heirs [17], specialist-trained civilians [18], paid physiotherapists [19] as well as female patients who were HIV positive [20]. In this study, the selected helpers or guides consisted of healthy older adults in residential aged care residents and were also referred to as Buddies, while individuals who have more significant disabilities–cognitive or physical or both–were referred to as older adults.

Several issues arose during the training period when conducting a buddy program, such as an increase in depressive symptoms observed among buddies [16]. This may be due to a lack of training or skills required during the buddy program which causes the individual who is providing assistance to feel depressed. The problems and challenges faced by these buddies may be avoided if there is guidance that they can refer to when helping or handling older adults who have more significant disabilities in their daily lives. Therefore, the study aimed to identify the effectiveness of the buddy program training module to enhance the daily living functions, i.e., BADL and IADL and indirectly promote the social involvement of the elderly in daily life as well as the emotional support of the elderly. This module is also developed according to the suitability of those who will use the module; namely, the residents in the residential aged care homes.

## Materials and methods

### Study design

The present study was a third phase–the intervention phase–of the primary study conducted between 27 August 2019 and 22 October 2019 for eight weeks. The proposed study was designed as a quasi-experimental study. This type of design was chosen to get two comparable groups of participants in the intervention group and control group. This quasi-experimental

study involved the comparison of the intervention group and control group without random selection [21–23]. The intervention group and control group of this study consisted of paired healthy older adults in residential aged care homes residents called "buddies" and older adults with more significant disabilities called "older adults".

This study has received ethical approval from UKM Research Ethics Committee (UKM PPI/111/8/JEP-2017-389).

## Participants

For the sample selection, in the first stage, the researcher selected two out of the 10 government residential aged care homes called 'Rumah Seri Kenangan" (RSK) by using simple random sampling. There are 10 RSK throughout Malaysia. Numbers representing each of the ten were marked on a small piece of paper, folded, and put in a small box. The selection was done by a researcher who was not involved in the study. The first draw from the box represented the intervention group and proved to be RSK Johor Bahru, while the second draw taken was RSK Malacca and represented the control group.

RSK Johor Bahru is located in a suburban area, but relatively close to the city of Johor Bahru. The number of residents is approximately 270. RSK Malacca is also located in a sub-town near city of Cheng, Malacca and the number of residents is about 300. Similarly, both RSK consist of residents from various races, ages, religions, and backgrounds. Next, participants from both groups underwent a screening test using Mini Mental State Examination (MMSE) to determine whether they met the pre-determined inclusion or exclusion criteria [24]. Then, participants were selected by purposive sampling according to their residential block to matched the buddies and older adults.

G-Power was used to determine the sample size for this study based on the study done by [16]. The total sample size was 44 pairs for both groups (including a 10% non-response bias). However, only 30 pairs of buddies and older adults participated in this study for both the experiment group and control group. This is due to the availability of the participant in each residential aged care home who were eligible to take part in this study based on the inclusion and exclusion criteria of the study.

The inclusion and exclusion criteria for the participants in the experimental group are listed to determine the subjects who are evaluated; the buddy program training module was given to the buddies throughout the intervention study while inclusion criteria for the control group are listed to meet the needs and requirements for analysis. The following are the two criteria used to determine the participants who are evaluated and involved in the intervention for this study as 'guiding buddies' and also those older adults who became the 'guided friends'.

The inclusion criteria for older adults as 'guiding buddies' were residents who (1) were aged 60 years and above; (ii) resided in residential aged care homes for more than 6 months; (iii) had intact cognitive functioning (Mini-Mental State Examination (MMSE) score above 19 marks) [24]; (iv) were physically intact (being independent and not using assistive devices); (v) could cooperate and were willing to assist the other residents; and (vi) had the ability to read and communicate verbally in the Malay language.

The inclusion criteria for older adults as 'guided friends' were residents who (i) are aged 60 years and above; (ii) reside in residential aged care homes for more than 6 months; (iii) have impaired cognitive functioning [Mini-Mental State Examination (MMSE) score below 19 marks] [24] and/or are physically impaired (need assistance in activities of daily living/ use aids to mobilize); are willing to participate in the study; (iv) and have the ability to read and communicate verbally in the Malay language.

## Instruments

A sociodemographic form was developed to obtain background information on the residents in the two residential aged care homes. The information contained in this form includes gender, age, race, religion, marital status, level of education, and length of stay.

The Modified Barthel Index (MBI) Malay version [9] was used in this study to measure the level of independence in activities of daily living demonstrated by the older adults. This form includes 10 items; namely, personal hygiene, bathing, toileting, climbing stairs, dressing, feeding, bladder control, bowel control, transfer and ambulation/wheelchair [9]. Each activity on this 10-item scale contains one to five levels of dependency ranging from 0 (cannot perform activities) to 2, 5, 10 and 15 (fully independent). Each score in all activities will be added to get the total score between 0 and 100. A higher score indicates a high degree of independence and vice versa. This form can be evaluated through observation of the respondent, interview with the respondent, and joint interview with the guardian. It will take approximately 10 minutes. However, it will require a longer duration if the researcher wants to make observations while the activities are being carried out by the respondent.

The Malay version of Performance Assessment of Self-Care Skills (PASS-MV) [11] is a performance-based evaluation form, determination based on criteria (individuals are evaluated according to performance based on established criteria) and observational assessments used by health practitioners to identify performance levels and changes in the functioning of daily life. This evaluation form consists of twenty-six tasks in total: five tasks for movement functions, three tasks for personal care, fourteen tasks for instrumental daily activities with cognitive emphasis and four instrumental daily activity assignments with physical emphasis. PASS has two versions: namely, the clinic version and the home version. Both versions are the same except for the materials used for some tasks. For example, for medication management, in the clinic version, there is a prescription label provided, but in the home version the client needs to use their own medicine. For the purpose of this study, the home version was chosen because it is compatible with the context of the participants involved in the study, namely the older adults in two residential aged care homes.

The activities contained in this assessment form are carried out with verbal instructions and according to the suitability of activities. While respondents carried out activities, researchers provided assistance when needed, starting with minimum help until assistance was required frequently. The stages of assistance are (1) Verbal supportive (encouragement); (2) Verbal Non-Directive (3) Verbal Directive; (4) Gestures; (5) Tasks of Environment Rearrangement; (6) Demonstration; (7) Physical guidance; (8) Physical support, and (9) Total assist. The original English version of the PASS form has high validity and reliability. The results of the exploratory factor analysis (EFA) test on all three domains, namely, independence, safety and adequacy, show that there is a dominant construct that meets the unidimensionality assumption for each construct. The Rasch analysis test showed that there were several tasks in PASS that had differences in difficulty for female older adults with depression living in wards compared to outpatient older adults. The Malay version of PASS-MV has very high content validity, for the content between items, the average I-CVI = 0.99 and the overall agreement by the evaluator S-CVI/UA = 0.926 [25].

Canadian Occupational Performance Measure (COPM) [26] is an individual assessment tool used to observe changes in self-perception in the client's achievements and satisfaction through problems identified in daily life. It is evaluated through a semi-structured interview between the interviewer and the client of between 20 minutes and 60 minutes, depending on the client's condition at the time. This evaluation tool tests five main problems that clients face in performing activities of daily living. This problem is determined by a 10-point scale which

will help in providing guidance for recovery. This assessment includes five steps to assist in interviewing clients on a semi-structured basis. The first step is that the client identifies problems in the performance of daily life. The second step is for the client to determine the most important problem by assessing the importance of each problem using a 10-point scale (from very unimportant to very important). The third step is to identify the five most important problems they face during their daily activities. The fourth step has clients assess their performance using a 10-point scale (from incapable to very capable of doing the task). In the fifth step the client has to assess self-satisfaction (from highly dissatisfied to very satisfied) for each of the five main problems identified.

This COPM scale was chosen because it proved reliability and validity. In the original study, Law et al. (1991) found that the reliability of Intraclass correlation coefficients for the average reading of performance was 0.67 (95% confidence interval). While reliability for satisfaction is 0.69 (95% CI). In this study, the COPM which had been translated from English to Malay in the last study will be used [27].

Bahasa Malaysia Depression Anxiety Stress Scale 21 (BM DASS21) [28] is a set of three self-report scales designed to measure the emotional states of depression, anxiety and stress. When completing this form, respondents (both buddies and older adults) need to state the symptoms they had experienced in the past week. Each item uses four scoring scales from 0 (not directly describing my situation) to 3 (very much or very often describing my situation). Reliability tests showed alpha Cronbach values were 0.75, 0.74 and 0.79 for depression, anxiety and stress, respectively, while the construct validity value contained the loading factor of 17 out of 21 items (0.31 to 0.75).

## Study procedure

The information about the study was displayed on the residential aged care homes' notice boards. This included the title of the study, inclusion and exclusion criteria, and how to contact the researchers if the residents were interested in participating in the study. The researcher was then provided the explanatory statement and consent form to the potential participants. Participants who agreed to participate in the study, they provided their informed consent by directly returning their signed forms to the researchers. For the participants who were unable to read, the impartial witness was present during the entire informed consent discussion.

## Statistical analysis

The Split-plot analysis of variance (SPANOVA) test was carried out to differentiate changes in the daily life function (basic daily activities and instrumental daily activities), social participation and emotional status between buddies in the experimental group and the control group. In this study, differences in the emotional status of the older adults between the experimental group and the control group were carried out. Data normality for each dependent variable (MBI, PASS-MV, COPM, BM DASS21) regarding each group and time was evaluated separately using the SPANOVA test. Samples of small sizes will be more sensitive to isolated objects (outliers) and cause abnormal data. Therefore, in addition to relying on the normality of the univariate, Cook's distance can be used to check the influence of the entire score on the results obtained. The value of Cook's distance for each participant's score of no more than 1 indicates no isolated multivariate (outliers) [29]. In addition, the assumption of variance homogeneity among the sample factors in SPANOVA was considered unviolated when the Levene test was not significant. The study synonymity for all tests is set at $p < 0.05$. The effect size (effect size) for SPANOVA (magnitude effect) is based on the partial values of eta squared (p2): 0.1 (good), 0.059 (medium) and 0.138 (strong) [30].

## Result

The buddy program training module involved 30 buddies and 30 older adults who needed help in their daily lives. Participants were divided into two groups:15 pairs in the experimental group and 15 pairs in the control group. Referring to Table 1, there was no significant difference in the characteristics of participants between the two groups ($p > 0.05$) except for the age of participants ($p < 0.05$).

The participants in the experimental group, who used training modules, had an average age of 66.67 years, while the participants in the control group had an average of 72.00 years. Both the experimental group and the control group had an equal number of participants, consisting of 8 females and 7 males. In both groups, the number of participants belonging to the Malay

**Table 1. Frequency differences (%) of socio-demographic characteristics among older adults.**

| | Mean±standard deviation[a] /Frequency(%) | | p-value |
|---|---|---|---|
| | Experiment group (n = 15) | Control group (n = 15) | |
| Age[a] | 66.67±3.309 | 72.00±4.660 | 0.001[c] |
| Gender[b] | | | 1.000[e] |
| Female | 8(53.3) | 8(53.3) | |
| Male | 7(46.7) | 7(46.7) | |
| Race[b] | | | 0.686[f] |
| Malay | 11(73.3) | 11(73.3) | |
| Chinese | 2(13.3) | 1(6.7) | |
| Indian | 2(13.3) | 1(6.7) | |
| Other | | 2(13.3) | |
| Religion[b] | | | 1.000[f] |
| Muslim | 12(80.0) | 12(80.0) | |
| Buddha | 1(6.7) | 1(6.7) | |
| Hindu | 2(13.3) | 1(6.7) | |
| Other | | 1(6.7) | |
| Marital status[b] | | | 0.413[f] |
| Single | 4(26.7) | 4(26.7) | |
| Married | 2(13.3) | 2(13.3) | |
| Widowed | 2(13.3) | 6(40.0) | |
| Widower | 7(46.7) | 3(20.0) | |
| Level of education[b] | | | 0.154[f] |
| No schooling | | 1(6.7) | |
| Primary school | 8(53.3) | 12(80.0) | |
| Secondary school | 6(40.0) | 2(13.3) | |
| College | 1(6.7) | | |
| Duration staying at aged care homes[b] | | | 0.805[f] |
| Less than 3 years | 7(46.7) | 6(40.0) | |
| 3 to 6 years | 3(20.0) | 5(33.3) | |
| 6 years and above | 5(33.3) | 4(26.7) | |

[a]Mean ± standard deviation.

[b]Frequency (%).

[c]Independent t-test.

[e]Chi-square.

[f]Fisher exact test.

race was higher compared to other races, with a total of 11 participants. Additionally, the Islamic religion had the largest representation among participants, with a total of 12 individuals. Both groups had an equal number of participants who were single (26.7%) and married (13.3%) and the remaining participants included individuals who were widows and widowers. The primary school level of education had the highest number with 20 participants in both groups and there were 13 participants who had resided in the RSK for less than three years.

There was no significant difference in the pretest between the experimental group and the control group for any of the variables (Table 2), Modified Barthel Index, Bahasa Malaysia Depression Anxiety Stress Scale 21-item (BM-DASS21), Malay version of Performance Assessment of Self-care Skills (PASS-MV) and Canadian Occupational Performance Measure (COPM).

A total of 30 buddies agreed to participate in the study which consisted of an experimental group and a control group as shown in Table 3. There was no significant difference for all demographic characteristics of the buddies ($p > 0.05$) where the mean age was 67.73 and 69.60 for the experimental group and control group, respectively. Both the experimental and control groups consisted of an equal number of male and female participants. However, there was s higher representation of Malays and Muslims compared to other races and religions, with 11 participants in the experimental group and 10 participants in the control group. There was also the same number of single participants in both groups; however, there were married participants in the control group (13.3%). The majority of participants from both groups had a primary school education level and a total of 12 participants had been living in residential aged care homes for more than six years. For BM DASS21 scores, there was no significant difference for the three variables studied: namely, depression, anxiety and stress.

**Table 2. The mean score of the pretest between the experimental group and control group for older adults.**

| Variable | Mean ± standard deviation[a] / Frequency (%)[b] / Median (IQR) | | p-value |
|---|---|---|---|
| | Experiment group (n = 15) | Control group (n = 15) | |
| Modified Barthel Index | 87.13±12.165 | 79.73±10.886 | 0.090[c] |
| BM DASS21: Depression | 1.67±1.759 | 1.80±1.320 | 0.816[c] |
| BM DASS21: Anxiety | 1.07±1.534 | 1.87±1.407 | 0.148[c] |
| BM DASS21: Stress | 2.13±2.588 | 1.67±1.291 | 0.539[c] |
| PASS-MV: Task 1: Bed mobility | 0.00(0) | 0.00(16) | 0.089[d] |
| PASS-MV: Task 2: Stair use | 0.00(1) | 0.00(27) | 0.595[d] |
| PASS-MV: Task 3: Toilet mobility and management | 0.00(0) | 0.00(3) | 0.595[d] |
| PASS-MV: Task 4: Oral hygiene | 0.00(0) | 0.00(0) | 1.000[d] |
| PASS-MV: Task 5: Bathtub and shower mobility | 0.00(0) | - | 0.775[d] |
| PASS-MV: Task 6: Trimming toenails | 0.00(0) | 0.00(36) | 0.217[d] |
| PASS-MV: Task 7: Dressing | 0.00(0) | 0.00(0) | 0.967[d] |
| PASS-MV: Task 8: Hijab wearing | 0.00(0) | - | 0.775[d] |
| PASS-MV: Task 9: Sarung wearing | 0.00(0) | 0.00(0) | 1.000[d] |
| PASS-MV: Task 15: Changing bed linens | - | 0.00(0) | 0.367[d] |
| PASS-MV: Task 20: Indoor walking | 0.00(0) | 0.00(0) | 0.806[d] |
| Canadian Occupational Performance Measure (COPM) | 12.93±1.624 | 11.93±2.576 | 0.214[c] |

[a]Mean ± standard deviation.

[b]Frequency (%).

[c]Independent t-test.

[d]Mann-Whitney U test.

**Table 3. Frequency differences (%) of socio-demography characteristics and pre-tests between the experimental group and control group among buddies.**

| Variable | Mean ± standard deviation[a] / Frequency (%)[b] / Median (IQR) | | p-value |
|---|---|---|---|
| | Experiment group (n = 15) | Control group (n = 15) | |
| Age[a] | 67.73±7.787 | 69.60±3.996 | 0.416[c] |
| Gender[b] | | | 1.000[e] |
| Female | 8(53.3) | 8(53.3) | |
| Male | 7(46.7) | 7(46.7) | |
| Race[b] | | | 0.436[e] |
| Malay | 11(73.3) | 10(66.7) | |
| Chinese | 3(20.0) | 1(6.7) | |
| Indian | 1(6.7) | 3(20.0) | |
| Others | | 1(6.7) | |
| Religion[b] | | | 0.109[f] |
| Muslim | 12(80.0) | 13(86.7) | |
| Buddha | 3(20.0) | | |
| Hindu | | 2(13.3) | |
| Marital status[b] | | | 0.517[f] |
| Single | 5(33.3) | 4(26.7) | |
| Married | | 2(13.3) | |
| Widowed | 2(13.3) | 4(26.7) | |
| Widower | 8(53.3) | 5(33.3) | |
| Level of education[b] | | | 0.520[f] |
| No schooling | 5(33.3) | 2(13.3) | |
| Primary school | 6(40.0) | 8(53.3) | |
| Secondary school | 4(26.7) | 5(33.3) | |
| Duration staying at aged care homes[b] | | | 0.386[f] |
| Less than 3 years | 3(20.0) | 5(33.3) | |
| 3 to 6 years | 7(46.7) | 3(20.0) | |
| 6 years and above | 5(33.3) | 7(46.7) | |
| Depression | | | |
| BM DASS21: | 0.47±0.516 | 0.73±1.163 | 0.427[c] |
| BM DASS21: Anxiety | 0.00(1) | 1.00(1) | 0.098[d] |
| BM DASS21: Stress | 1.40±1.844 | 1.20±1.612 | 0.754[c] |

[a]Mean ± standard deviation

[b]Frequency (%).

[c]Independent t-test.

[d]Mann-Whitney U test.

[e]Chi square test.

[f]Fisher exact test.

## Basic activities of daily living (BADL)

The effectiveness of the buddy program training module for daily living function among older adults was evaluated using the Modified Barthel Index (MBI) form–Malay version. The difference between older adults in the experimental group and the control group was based on the overall score of all items in the MBI.

Table 4. Comparison of experimental and control groups with older adults in basic activities of daily living (BADL).

| Group | Time | | Within group | | Between group | | Interaction | |
|---|---|---|---|---|---|---|---|---|
| | Pre | Post | $p$ | $\eta p^2$ | $p$ | $\eta p^2$ | $p$ | $\eta p^2$ |
| Experiment | 87.13±12.17 | 89.27±10.00 | 0.010* | 0.215 | 0.076 | 0.108 | 0.735 | 0.004 |
| Control | 79.73±10.89 | 82.47±10.07 | | | | | | |

The findings showed that there was a significant difference in the main effect of time F (1, 28) = 7.67, $p$ = 0.010, $\eta p^2$ = 0.215, where the MBI score after using the buddy program training module for older adults (M = 85.87, SD = 10.45) was higher than before using the training module (M = 83.43, SD = 11.95). There was no significant difference in the main effects of group F (1, 28) = 3.40, $p$ = 0.076, $\eta p^2$ = 0.108. There was no significant difference in the effect of the interaction F (1, 28) = 0.117, $p$ = 0.735, $\eta p^2$ = 0.004. Refer to Table 4.

## Instrumental activities of daily living (IADL)

The effectiveness of the buddy program training module for older adults on the performance in instrumental daily activities by using the Malay version of the Performance Assessment of Self-care Skills (PASS-MV) form.

The assessment of instrumental activities of daily living among older adults in the experiment and control groups is based on the total score for each item in the PASS-MV (Table 5). There are only 11 items out of 26 items that are relevant for analysis. These items are Task 1 (Bed mobility), Task 2 (Stair use), Task 3 (Toilet mobility and management), Task 4 (Oral

Table 5. Comparison of experimental and control groups with older adults in instrumental activities of daily living (IADL).

| Item | Group | Time | | Within group | | Between group | | Interaction | |
|---|---|---|---|---|---|---|---|---|---|
| | | Pre | Post | $p$ | $\eta p^2$ | $p$ | $\eta p^2$ | $p$ | $\eta p^2$ |
| Bed mobility | Exp | 0.67±1.92 | 0.47±1.25 | 0.273 | 0.043 | 0.025* | 0.167 | 0.431 | 0.022 |
| | Ctrl | 5.87±8.15 | 4.67±7.65 | | | | | | |
| Stair use | Exp | 3.73±8.92 | 2.07±6.26 | 0.198 | 0.058 | 0.186 | 0.062 | 0.832 | 0.002 |
| | Ctrl | 8.40±12.48 | 7.20±12.40 | | | | | | |
| Toilet mobility and management | Exp | 4.13±11.95 | 3.73±10.66 | 0.562 | 0.012 | 0.629 | 0.008 | 0.250 | 0.047 |
| | Ctrl | 2.33±4.53 | 2.47±4.55 | | | | | | |
| Trimming toenails | Exp | 4.80±12.67 | 7.20±14.91 | 0.334 | 0.033 | 0.088 | 0.101 | 1.000 | 0.000 |
| | Ctrl | 14.40±18.26 | 16.80±18.59 | | | | | | |
| Dressing | Exp | 2.67±7.20 | 2.67±7.20 | 0.461 | 0.020 | 0.932 | 0.000 | 0.461 | 0.020 |
| | Ctrl | 2.13±5.63 | 2.80±5.81 | | | | | | |
| Hijab wearing | Exp | 0.13±0.52 | 0.07±0.26 | 0.326 | 0.034 | 0.326 | 0.034 | 0.326 | 0.034 |
| | Ctrl | 0.00±0.00 | 0.00±0.00 | | | | | | |
| Wearing and tying a sarong | Exp | 1.07±4.13 | 1.00±3.87 | 0.427 | 0.023 | 0.903 | 0.001 | 0.971 | 0.333 |
| | Ctrl | 0.87±3.36 | 1.53±4.05 | | | | | | |
| Changing bed linens | Exp | 0.00±0.00 | 0.00±0.00 | 0.326 | 0.034 | 0.129 | 0.080 | 0.326 | 0.034 |
| | Ctrl | 7.67±18.90 | 7.60±18.90 | | | | | | |
| Indoor walking | Exp | 1.60±6.20 | 1.60±6.20 | 0.326 | 0.034 | 0.963 | 0.000 | 0.326 | 0.034 |
| | Ctrl | 2.00±5.75 | 1.40±5.42 | | | | | | |

Exp = Experiment.

Ctrl = Control.

hygiene), Task 5 (Bathtub and shower mobility), Task 6 (Trimming toenails), Task 7 (Dressing), Task 8 (Hijab wearing), Task 9 (Wearing and tying a sarong), Task 15 (Changing bed linens) and Task 20 (Indoor walking). The SPANOVA test was conducted to determine the effectiveness of the buddy program training module for older adults for both groups in the pre-test and post-test.

The findings showed no significant difference in the effect of time for all tasks in PASS-MV. However, there was a significant difference in the main effect of group F $(1, 28)$ = 5.60, $p = 0.025$, $\eta p^2 = 0.167$ where the bed mobility activity score (Task 1) of the control group (M = 5.27, SD = 7.79) was higher compared to the experimental group (M = 0.57, SD = 1.59). There was no significant difference in the effect of the interaction F $(1, 28)$ = 0.639, $p = 0.431$, $\eta p^2 = 0.022$. For Task 2, stair use, there was no significant difference in the effect of the interaction F $(1, 28)$ = 0.046, $p = 0.832$, $\eta p^2 = 0.002$. There was no significant difference in the main effects of time, group and interaction for the activity of changing bed linens, F $(1, 28)$ = 1.00, $p = 0.326$, $\eta p^2 = 0.034$, F $(1, 28)$ = 2.45, $p = 0.129$, $\eta p^2 = 0.080$ and F $(1, 28)$ = 1.00, $p = 0.326$, $\eta p^2 = 0.034$ respectively. There was no significant difference in the effect of interaction of other items found in the IADL form in this study.

## Social participation

The effectiveness of the buddy program training module for social participation among older adults by using the Malay version of the Canadian Occupational Performance Measure (COPM) form.

Social participation was evaluated in this study using COPM forms, which encompass overall scores comprising performance and satisfaction scores for daily living activities. SPANOVA tests were conducted to determine the effectiveness of the buddy program training module among older adults for both groups in pre-tests and post-tests.

The results showed that there was a significant difference in the main effect of time F $(1, 28)$ = 8.52, $p = 0.007$, $\eta p^2 = 0.233$ with social participation score after using the buddy program training module for older adults (M = 13.87, SD = 1.98) showing improvement compared to before the use of the training module (M = 12.43, SD = 2.18). There was no significant difference in the main effects of group F $(1, 28)$ = 1.47, $p = 0.237$, $\eta p^2 = 1.462$. There was no significant difference in the effect of the interaction F $(1, 28)$ = 0.373, $p = 0.546$, $\eta p^2 = 0.013$ (Refer to Table 6).

## Emotional status of the older adults

The effectiveness of the buddy program training module among older adults on the emotional status using the existing Malay version of the Bahasa Malaysia Depression Anxiety Stress (BM DASS21). The SPANOVA test was conducted to determine the effectiveness of the buddy program training module on the emotional status of older adults in experimental groups and control groups for pre-test and post-test. The score for this measurement was compared for 3 domains: depression, anxiety and stress.

**Table 6. Comparison of experimental and control groups among older adults in social participation.**

| Group | Time | | Within group | | Between group | | Interaction | |
|---|---|---|---|---|---|---|---|---|
| | Pre | Post | $p$ | $\eta p^2$ | $p$ | $\eta p^2$ | $p$ | $\eta p^2$ |
| Experiment | 12.93±1.62 | 14.07±1.44 | 0.007* | 0.233 | 0.237 | 0.050 | 0.546 | 0.013 |
| Control | 11.93±2.58 | 13.67±2.44 | | | | | | |

**Table 7. Comparison of experimental and control groups among older adults in emotional status.**

| Item | Group | Time | | Within group | | Between group | | Interaction | |
|------|-------|------|------|------|------|------|------|------|------|
| | | Pre | Post | $p$ | $\eta p^2$ | $p$ | $\eta p^2$ | $p$ | $\eta p^2$ |
| 1 | Exp | 1.67±1.78 | 1.13±1.73 | 0.221 | 0.053 | 0.498 | 0.017 | 0.459 | 0.020 |
| | Ctrl | 1.80±1.32 | 1.67±1.18 | | | | | | |
| 2 | Exp | 1.07±1.53 | 0.67±0.98 | 0.007* | 0.236 | 0.204 | 0.057 | 0.218 | 0.054 |
| | Ctrl | 1.87±1.41 | 0.87±0.92 | | | | | | |
| 3 | Exp | 2.13±2.59 | 1.07±1.58 | 0.058 | 0.122 | 0.734 | 0.004 | 0.446 | 0.021 |
| | Ctrl | 1.67±1.30 | 1.20±0.86 | | | | | | |

1 Depression.

2 Anxiety.

3 Stress.

Exp = Experiment.

Ctrl = Control.

The findings showed that there was a significant difference in the main effect of time F (1, 28) = 8.65, $p$ = 0.007, $\eta p^2$ = 0.236 with a lower anxiety score after using the buddy program training module for older adults (M = 0.77, SD = 0.935) than before using the training module (M = 1.47, SD = 1.50). There was no significant difference in the main effect of the group for depression scores F (1, 28) = 0.47, $p$ = 0.498, $\eta p^2$ = 0.017, anxiety scores F (1, 28) = 1.70, $p$ = 0.204, $\eta p^2$ = 0.057, and stress scores F (1, 28) = 0.12, $p$ = 0.734, $\eta p^2$ = 0.004. There was no significant difference in the effects of interaction on depression F (1, 28) = 0.565, $p$ = 0.459, $\eta p^2$ = 0.020. As for anxiety, there is a significant difference in the effect of the interaction F (1, 28) = 1.588, $p$ = 0.218, $\eta p^2$ = 0.054. For stress, there was no significant difference in the effect of the interaction F (1, 28) = 0.597, $p$ = 0.446, $\eta p^2$ = 0.021. Refer to Table 7 for the performance of the emotional status of older adults.

## Emotional status of the buddies

The effectiveness of the buddy program training module on the emotional status of buddies was measured using the existing Malay version of the Bahasa Malaysia Depression Anxiety Stress Scale 21-item (BM DASS21) form. The SPANOVA test was conducted to determine the effectiveness of the buddy program training module on the emotional status of buddies in experimental groups and control groups for pre-test and post-test. The score for this measurement was compared for 3 domains: depression, anxiety, and stress (Table 8).

There was a significant difference in the main effects of group F (1, 28) = 7.05, $p$ = 0.013, $\eta p^2$ = 0.201 with an increase in depression scores for the experimental group being lower after using the training module (M = 0.53, SD = 0.63) compared to the control group (M = 1.33, SD = 1.50). There were significant differences in the effects of interaction in depression F (1, 28) = 4.41, $p$ = 0.045, $\eta p^2$ = 0.136 in which the experimental group showed lower score increases after using the module (M = 0.60, SD = 0.74) and before using the module (M = 0.47, SD = 0.516), compared with the control group that carried out daily activities as usual after (M = 1.93, SD = 1.58) and before the intervention was carried out (M = 0.73, SD = 1.16). As for anxiety, there was no significant difference in the effect of the interaction F (1, 28) = 0.477, $p$ = 0.496, $\eta p^2$ = 0.017. Similarly, for stress, there was no significant difference in the effect of the interaction F (1, 28) = 0.679, $p$ = 0.417, $\eta p^2$ = 0.024.

**Table 8. Comparison of experimental and control groups among buddies in emotional status.**

| | Group | Time | | Within group | | Between group | | Interaction | |
|---|---|---|---|---|---|---|---|---|---|
| | | Pre | Post | $p$ | $\eta p^2$ | $p$ | $\eta p^2$ | $p$ | $\eta p^2$ |
| 1 | Exp | 0.47±0.52 | 0.60±0.74 | 0.014* | 0.198 | 0.013* | 0.201 | 0.045* | 0.136 |
| | Ctrl | 0.73±1.16 | 1.93±1.58 | | | | | | |
| 2 | Exp | 0.33±0.49 | 0.40±0.83 | 0.733 | 0.004 | 0.091 | 0.098 | 0.496 | 0.017 |
| | Ctrl | 0.80±0.78 | 0.60±0.83 | | | | | | |
| 3 | Exp | 1.40±1.85 | 1.13±1.19 | 0.907 | 0.000 | 0.944 | 0.000 | 0.417 | 0.024 |
| | Ctrl | 1.20±1.61 | 1.40±1.24 | | | | | | |

1 Depression.

2 Anxiety.

3 Stress.

Exp = Experiment.

Ctrl = Control.

## Discussion

### The effectiveness of the buddy program training module between the experimental group and control group among older adults in BADL

The main purpose of this study is to examine the development and effectiveness of the buddy program training modules among older adults in improving their daily living functions in BADL. This training module has been developed for buddies in this study to assist older adults with more significant disabilities–either cognitive, physical, or both.

It was found that there was a significant increase in the BADL score among older adults after the intervention was carried out between buddies and older adults by using the buddy program training module for the experimental group and normal daily activities for the control group. However, there was no significant difference in the effect of interaction between the experimental group and the control group.

The intervention did not have a fixed number of meetings between the buddies and older adults when considering the overall frequency of assistance provided by the buddy upon reevaluation. The buddies have the flexibility to assist older adults in daily living activities at their own convenience, choosing activities and times that suit their comfort level. However, they were briefed during the demonstration session on how to carry out all the activities in the buddy program training module prior to the eight weeks' intervention. The findings of this study are in line with a study conducted by [16] that also used a buddy system to improve the well-being of older adults in aged care homes. The results of this past study indicated that older adults with dementia did not show a significant difference from the control group for all measurements used.

However, a quasi-experimental study conducted on 21 experimental groups and 21 control groups showed that there was a significant increase in feeding, dressing and personal hygiene activities for older adults who received the Enhanced Effectiveness and Care Program (SCSEEP) [31]. Studies conducted on home-dwelling older adults involved in home program interventions had low functional deterioration based on disability loss scores [32], and, based on this study, also showed that the increase in ADL scores for the experimental group was almost the same as for the control group after the intervention. This finding could be due to the Hawthorne effect where the tendency of older adults who are respondents is to change or improve their behavior when the assessment is being carried out [33].

## The difference in the effectiveness of the buddy program training module between the experimental group and control group among older adults in IADL

Based on the findings from the PASS-MV form, there was no significant difference in the instrumental activities of daily living of changing bed sheets. This shows that the assistance provided by the buddies to the older adults who need help in their daily lives by using this training module for the older adults has no effect on this housework activity. Upon reviewing the situation at the residential aged care homes, it was found that certain older adults residing in the wards do not require the bed sheet to be fitted. This depends on the health condition of the older adults where, if they experience problems with defecation and urination, they will be placed in a bed with a disposable waterproof sheet to facilitate the cleaning process if they are unable to go to the bathroom immediately. Furthermore, within the wards where older adults require assistance, there are other older adults who are healthy and capable of taking care of themselves. These residents voluntarily help the other residents, by changing their bed sheets on a daily basis. However, this leads to reliance on the healthier older adults to carry out activities. Studies conducted in Malaysia on the prevalence of daily activities among seniors in four districts in Selangor indicated that good social support from good partners is associated with a high risk of dependency [7].

Another factor that may have contributed to the findings of this study was that the buddies did not receive enough training before the intervention was carried out for the experimental group, which is the group that used the buddy program training module. In this study, the buddies were given two demonstration sessions (1 hour per session) for all the activities contained in the module. Previous studies were also seen to be in line with this study, where training, communication, and insufficient feedback between the buddy partners were among the negative issues that arose during the mentoring program [19]. Nevertheless, there are other studies that show that the level of adherence of older adults guided by people who do not have a high level of education, is not much lower than that from care delivered by nurses trained in physical activity and functionality [34].

## The difference in the effectiveness of the buddy program training module between the experimental group and control group among older adults in social participation

Social participation was found to be associated with good functional skills, well-being, and quality of life; it is considered an important assessment to be conducted in the care of older adults [12]. The results showed that there was a significant increase in the main effect of time on social participation scores compared to before the module was used. However, the results showed that there was no significant difference in the effect of interaction on social participation scores among older adults after using modules for experimental groups and control groups.

The results of this study are based on the findings from the activities carried out by the buddies and the older adults during the intervention. The lack of frequency of activities carried out between buddies and older adults may affect the social participation of older adults in their daily lives involving BADL and IADL. This is because individuals who engage in active activities such as going out and carrying out outdoor activities have a high sense of life satisfaction, where social contact is found to contribute to that sense of life satisfaction [35]. The frequency of assistance by the buddies can also be influenced by the activities of older adults along with the other group of people from external organizations or institutions that come daily to carry

out other activities. Due to the other activities that occurred that were not related to this study, the time required to carry out the intervention within the specified period was interrupted.

The development of this new module is crucial because it can increase motivation and attract interest, and, most importantly, can encourage involvement in loaded activities and generate excitement. In addition, it has been found that social networks and active social participation of older adults can reduce the process of memory impairment [36–38]. This mentoring training module for the older adults was developed to provide guidance to the buddies to assist those older adults with more significant disabilities, who need help in BADL and IADL, while, at the same time, being indirectly aimed at increasing the social participation of the elderly in daily life. However, the results of this study do not support this purpose, as the activities are focused on activities that involve the function of daily living only. Furthermore, relaxing activities such as games or recreation can be included in the module to encourage the elderly to increase their involvement in an activity.

## Emotional status of the older adults between the experimental group and control group

Based on the analysis tests conducted, there was a significant difference in the main effect of time in anxiety scores; there was a reduction in scores after using the training module compared to before using the module. However, the results showed no significant difference in the effects of interactions between the three domains in DASS21: namely, depression, anxiety and stress for the experimental group and the control group.

These results showed that the BADL and IADL activities that were carried out using the training modules for the older adults did not indirectly reduce the poor emotional status of the elderly in terms of depression, anxiety and stress during activities. However, the results of this study can be used in developing other strategies that can be carried out to improve the emotional status of older adults by involving other activities that are more appropriate in encouraging their participation. There is no doubt that the importance of appropriate activities can encourage engagement and in turn can overcome the negative emotional status of the elderly. A study conducted by Husaini and colleagues [39] found that group therapy conducted for female older adults with depression could reduce the feeling of depression among Caucasian older adults in the age range from 55 years to 75 years. In addition, other research results also found that there was a significant reduction in depression, anxiety and anxiety among older adults who attended a mindfulness training program [40].

Although this study did not demonstrate the effectiveness of this module on older adults for all dependent variables (BADL, IADL, social participation and emotional status), based on anecdotal records, according to the buddies, older adults had increased some activities in BADL, such as the dressing activity. According to an interview conducted informally with the buddies after the post-test, there were older adults who were unable to don their own clothes initially and needed maximal assistance from their buddies. However, from one to two weeks after the intervention, these older adults were able to don clothes on their own with minimal supervision from the buddies.

## Emotional status of the buddies between experimental group and control group

The role of a caregiver is very important. The effects of care can directly affect emotions, stress, amount of sleep, and physical problems such as body pain from moving older adults who need help in their daily lives. There are also other factors that affect the ability of the buddies to

provide optimal assistance to the elderly such as health disorders, weakness, tiredness, lethargy, and even the attitudes of older adults who receive the assistance.

The results also showed that there was a significant major effect of depression among buddies after using the training module for the older adults. In addition, the findings showed that there was a significant interaction effect in which depression scores increased; however, depression scores in the experimental group were no different to those of the control group and were found to show a significant improvement after the intervention was carried out. These findings show that the use of this module does affect the emotional status of the buddies while providing assistance to the older adults who need help in their daily lives. This may be due to other factors that affect the emotions of the older adults residing in the residential aged care homes.

Studies on the relationship between the burden carried by older adults caregivers and the symptoms of anxiety remain unclear although past studies have suggested that caregiver burden is an early reaction when there is a need for care that can result in the appearance of symptoms of anxiety [41]. Caregivers need to carry out the duty of care for the older adults who need help in daily life and at the same time need to adapt to their own changed daily life routine as usual. The burden borne by older adult caregivers can be detrimental to the well-being or emotions of caregivers due to the duty of care they must face [42]. Nurul Hudani et al. [43] also stressed that emotional support and social support are important elements for those in the role of carer of older adults, as failure to obtain the necessary support can lead to depression and disrupt their well-being.

In addition, one of the main issues that can contribute to depression among these buddies is health factors. Buddies are themselves older adults residing in the same residential aged care homes as older adults who need help in their daily lives. Although the buddy might be in good health, nevertheless, the responsibility of helping the older adults has, to some extent, disturbed their emotions when they must do it alone. Those who may have some health problems might need to overcome their own health problems while, at the same time, helping the older adults who need to be assisted; this can cause them depression. This finding is in line with a study conducted by Ahn and Kim [44], which determined that among the resulting themes were the need for care involving quality care, difficulties in care, lethargy, encumbrance, need for care, and appreciation for the help provided. According to the study, too many difficulties associated with care duties caused the helper to feel lethargic, physically ill, and also to feel helpless.

Following the implementation of this buddy program training module, the depression scores of the two groups showed no significant differences, but there was a significant improvement in the control group after the intervention was carried out. Based on informal interviews conducted with the buddies after the post-test, the anecdotal record shows that the guides felt obliged to carry out joint activities with older adults who need help in everyday life. This training module for guides is also easy for the guide to follow and understand.

## Limitations and recommendations for future research

The buddy program training module was developed in the Malay language only. This module can be translated into other languages such as Mandarin or Tamil to be used by the other races in Malaysia. The activities contained in this module focus on BADL and IADL only. It is suggested that future studies accommodate other activities that are appropriate for older adults such as leisure activities. The buddy program training module can be enhanced by incorporating game activities conducted in pairs or groups, allowing for an evaluation of the effectiveness of different activities in improving the daily live activities of older adults. In addition, future

studies are also suggested where the improved modules should include activities related to breathing techniques to reduce the stress of the buddies while using the module. The study also focused on older adults residing in residential aged care homes only. Future studies could look at using this module for community-dwelling older adults and their family members, neighbours, volunteers, and friends who can act as buddies. This module is deemed suitable for use by any such individuals who are dealing with older adults, as it encompasses daily activities and instrumental daily activities that older adults typically engage in during their daily lives.

## Conclusion

The present study evaluates the effectiveness of the buddy program training module to enhance the daily living function (BADL and IADL), social participation, and emotional status of older adults in residential aged care homes. There were no significant interaction effects of the experiment group and control group on BADL, IADL, social participation, and emotional status. However, this module is seen to encourage older adults to be independent in BADL (don clothes) based on informal interviews with buddies. Therefore, this buddy program training module can be used as a guideline or reference in managing older adults with more significant disabilities in their daily living.

## Acknowledgments

The authors would like to thank all participants involved in the intervention phase.

## Author Contributions

**Formal analysis:** Siti Noraini Asmuri.

**Investigation:** Siti Noraini Asmuri.

**Methodology:** Siti Noraini Asmuri, Hanif Farhan Mohd Rasdi.

**Resources:** Siti Noraini Asmuri.

**Supervision:** Masne Kadar, Nor Afifi Razaob, Chai Siaw Chui, Hanif Farhan Mohd Rasdi.

**Validation:** Siti Noraini Asmuri, Masne Kadar, Nor Afifi Razaob, Chai Siaw Chui, Hanif Farhan Mohd Rasdi.

**Writing – original draft:** Siti Noraini Asmuri.

**Writing – review & editing:** Siti Noraini Asmuri, Masne Kadar, Nor Afifi Razaob.

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
