## [Decision Letter · Decision Letter 0]

12 Dec 2023

PONE-D-23-26893The effectiveness of the buddy program training module to enhance the daily living function, social participation and emotional status of older adults in residential aged care homesPLOS ONE

Dear Dr. asmuri,

Thank you for submitting your manuscript to PLOS ONE. After careful consideration, we feel that it has merit but does not fully meet PLOS ONE’s publication criteria as it currently stands. Therefore, we invite you to submit a revised version of the manuscript that addresses the points raised during the review process.

We look forward to receiving your revised manuscript.

Kind regards,

Adegoke Adefolalu, MBChB, MPH, PhD, FRSPH

Academic Editor

PLOS ONE

Reviewers' comments:

Reviewer's Responses to Questions

**Comments to the Author**

1. Is the manuscript technically sound, and do the data support the conclusions?

Reviewer #1: Yes

Reviewer #2: Yes

2. Has the statistical analysis been performed appropriately and rigorously? 

Reviewer #1: Yes

Reviewer #2: Yes

3. Have the authors made all data underlying the findings in their manuscript fully available?

Reviewer #1: Yes

Reviewer #2: Yes

4. Is the manuscript presented in an intelligible fashion and written in standard English?

Reviewer #1: Yes

Reviewer #2: Yes

5. Review Comments to the Author

Reviewer #1: The authors did a work worth publishing to an international audience and did a rigorous modern statistical analysis. However, questions about the methodology. The control group that were selected was based on a mini mental state of 19 and above, which supposes that they had mild to normal range of cognition. Also. the method of recruiting the participants presupposes adequate intellectual functioning and ability to read in Malay and understanding especially among the participants . Adverts on the notice board was the major method of information which means that they were voluntary , but questions remain about understanding of information presented and attention paid by the elderly to these notices in a rehab. If cognitive functioning is poor then understanding and informed consent may be doubtful as kins were not contacted. Additionally, the authors included all races but limited the restriction to verbally and readability in Malay language , only to suggest translations as recommendation ? Why were 2 out of 10 chosen randomly, that is 0.2 , within the sub region of similar social setting .i feel it was rather a convenience sample. The outlay of the residential groups should have been structured in such a way to represent more generalization.

The confounding factors are much and authors should note that even age and worsening dementia along with age may be a factor for differences in BADL. The quasi experimental was skewed to more of Malay residents and little representation of other groups and religion. How did they translate the instruments was there an effort to back translate in order to achieve more appropriate content validity and reliability.

Reviewer #2: Reviewer’s comments (Yetunde Adeniyi)

• Overall, it is a good paper that should be considered for publication. It emphasizes community participation and peer support.

• Under the inclusion criteria, the cut off mark on MMSE for both the guiding adults and guided adults were put at scores above 19. Authors should attend to this, the cut off points for both groups should not be the same.

• The authors should provide a summary of the study procedure, what we have in the paper right now are detailed description of the study instruments. There is a need to have a separate section for study procedures detailing what was done.

6. PLOS authors have the option to publish the peer review history of their article (what does this mean?). If published, this will include your full peer review and any attached files.

Reviewer #1: No

Reviewer #2: **Yes: **Dr Yetunde Adeniyi

---

## [Author Response · Author response to Decision Letter 0]

12 Mar 2024

Thank you for all the comments and suggestions. The manuscript has been addressed to improve readability.

---

## [Editor Report · Decision Letter 1]

18 Mar 2024

The effectiveness of the buddy program training module to enhance the daily living function, social participation and emotional status of older adults in residential aged care homes

PONE-D-23-26893R1

Dear Dr. Siti Noraini Asmuri,

We’re pleased to inform you that your manuscript has been judged scientifically suitable for publication and will be formally accepted for publication once it meets all outstanding technical requirements.

Kind regards,

Prof Adegoke Adefolalu, MBChB, MPH, PhD, FRSPH

Academic Editor

PLOS ONE
---

## [Editor Report · Acceptance letter]

25 Mar 2024

PONE-D-23-26893R1 

PLOS ONE

Dear Dr. asmuri, 

I'm pleased to inform you that your manuscript has been deemed suitable for publication in PLOS ONE. Congratulations! Your manuscript is now being handed over to our production team.

Kind regards, 

on behalf of

Prof Adegoke Adefolalu 

Academic Editor

PLOS ONE